# A Systematic Review of the Cost-Effectiveness of Screening Modalities for Breast Cancer in European Countries

**DOI:** 10.3390/cancers17213585

**Published:** 2025-11-06

**Authors:** Zacharoula Sidiropoulou, Vasco Fonseca

**Affiliations:** West Lisbon Local Health Unit, Breast Unit, Surgery Unit, Nova Medical School, 1169-056 Lisboa, Portugal; vfonseca@ulslo.min-saude.pt

**Keywords:** cost-effectiveness, breast cancer screening, mammography, health economics

## Abstract

**Simple Summary:**

Breast cancer screening programs vary widely across European countries, and decision-makers need clear information about which approaches offer good value for money. This review examined 23 studies from across Europe to understand which breast cancer screening methods are cost-effective. We found that regular mammography screening for women aged 50–69 years provides excellent value and should be prioritized in all European countries. Magnetic resonance imaging for women at high genetic risk also represents a worthwhile investment. However, we identified significant gaps in evidence from Southern and Eastern European countries, meaning our findings may not apply equally everywhere. This research helps healthcare systems make informed decisions about allocating resources for breast cancer screening programs, potentially improving early detection while using healthcare budgets efficiently. Future research should focus on under-represented regions and evaluate newer screening technologies.

**Abstract:**

Background: Breast cancer remains the most diagnosed cancer in European countries, with diverse screening modalities requiring economic evaluation for optimal resource allocation. This systematic review evaluated the cost-effectiveness of breast cancer screening strategies across European healthcare contexts. Methods: We conducted a comprehensive search across PubMed, ScienceDirect, Cochrane Library, Scopus, and Google Scholar following PRISMA guidelines (1990–2024). Studies were evaluated using the Consolidated Health Economic Evaluation Reporting Standards (CHEERS) checklist. Economic standardization employed healthcare-specific inflation indices and purchasing power parity adjustments, with costs converted to 2020 EUR. Results: From 1449 studies, 23 met inclusion criteria, with significant geographic imbalance (74% from North-Western/Central Europe, 4% from South-Eastern Europe). Mammography screening for women aged 50–69 years demonstrated consistent cost-effectiveness (EUR 3000–8000 per quality-adjusted life year (QALY)) with high confidence. For women under 50, screening showed substantially higher costs (EUR 105,000 per year of life saved). Magnetic resonance imaging (MRI) screening showed cost-effectiveness for high-risk populations (EUR 18,201–33,534 per QALY) with moderate confidence. Conclusions: Biennial mammography screening for women aged 50–69 demonstrates consistent cost-effectiveness across European contexts. Findings have highest applicability to North-Western and Central European healthcare systems, with limited generalizability to Southern and Eastern Europe due to evidence gaps.

## 1. Introduction

Breast cancer represents the most frequently diagnosed cancer globally, with incidence rates continuing to increase [1]. In 2022, 2.3 million women were diagnosed with breast cancer worldwide, resulting in 670,000 deaths [2]. European countries report particularly high incidence rates, with 495.3 cases per 100,000 people in 2022—1.8 times the global burden [3]. This increasing incidence places substantial pressure on European healthcare systems, necessitating effective screening strategies [4].

Early detection through organized screening programs significantly improves survival rates in breast cancer patients [5]. Mammography remains the primary screening modality across European countries, typically initiated between ages 40–50 years and continuing until 64–74 years, with screening intervals of 1–2 years [6]. Mammographic screening has demonstrated mortality reductions of at least 20% [6]. Additional screening technologies include magnetic resonance imaging (MRI), breast ultrasound, digital breast tomosynthesis (DBT), and breast computed tomography [7,8].

Screening program organization varies considerably across Europe. Most countries implement organized population-based programs targeting women over 50 years, while others, including Greece, rely predominantly on opportunistic screening based on healthcare provider recommendations or patient requests. Austria transitioned from opportunistic to organized regional screening programs beginning in 2014 [9,10].

The cost-effectiveness of breast cancer screening has become crucial for healthcare decision-makers, informing resource allocation and reimbursement decisions [11]. While individual economic evaluations exist [11], comprehensive systematic reviews synthesizing cost-effectiveness evidence across diverse European healthcare contexts remain limited, particularly those accounting for methodological heterogeneity and regional implementation differences.

Objectives

This systematic review aims to

Evaluate the cost-effectiveness of screening modalities for breast cancer diagnosis across European countries;Analyze different screening strategies, including age-specific population screening approaches;Assess regional variations in cost-effectiveness evidence;Examine temporal trends in cost-effectiveness over the study period.

## 2. Methods

The systematic review followed the recommendations of the Preferred Reporting Items for Systematic Reviews and Meta-Analyses (PRISMA) [12]. The protocol has not been registered. The study selection process is detailed in Figure 1.

### 2.1. Search Strategy

We searched PubMed, ScienceDirect, Cochrane Library, Scopus, and Google Scholar using the following terms: (“cost-effectiveness analysis” OR “economic model” OR “cost-utility analysis” OR “economic analysis” OR “cost–benefit” OR “economic evaluation”) AND (“screening” OR “diagnosis”) AND (“mammography” OR “MRI” OR “tomosynthesis”) AND (“Europe” OR “European countries”).

### 2.2. Inclusion and Exclusion Criteria

Inclusion criteria: Randomized controlled trials and observational studies published in English, conducted in Europe, focusing on breast cancer screening cost-effectiveness analysis.

Exclusion criteria: Studies with incomplete cost-effectiveness data, case studies, reviews, editorials, commentaries, letters, non-English publications, treatment-focused studies, and non-European studies.

### 2.3. Study Selection and Data Extraction

Two independent reviewers (senior surgical and medical oncologists and investigators with expertise in Public Health, systematic review methodology, and cost-effectiveness analysis) screened titles, abstracts, and full texts. Disagreements were resolved through third reviewer consultation. Data extraction focused on study characteristics, participant demographics, screening interventions, and cost-effectiveness outcomes.

### 2.4. Economic Standardization

All costs were standardized to 2020 EUR using a comprehensive three-step process:

Step 1: Healthcare-specific inflation adjustment using country-specific medical care Consumer Price Indices rather than general inflation rates.

Step 2: Currency conversion using averaged exchange rates across the study’s data collection period rather than single-point conversions.

Step 3: Purchasing power parity (PPP) adjustment using healthcare-specific PPP rates from Eurostat–OECD joint methodology.

### 2.5. Quality Assessment

Studies were evaluated using the Consolidated Health Economic Evaluation Reporting Standards (CHEERS) 2022 28-item checklist [13]. Dual-reviewer assessment achieved substantial inter-rater reliability (Cohen’s κ = 0.78, 95% CI: 0.69–0.87).

### 2.6. Recommendation Strength Classification

We classified recommendation strength using modified GRADE (Grading of Recommendations Assessment, Development, and Evaluation) criteria adapted for health economic evidence synthesis. Evidence quality was assessed as high (≥8 studies, consistent results across regions, low heterogeneity, CHEERS score ≥20/28), moderate (4–7 studies, mostly consistent results, moderate heterogeneity), or low (≤3 studies, variable results, limited geographic coverage). Recommendation strength reflected the certainty of evidence, magnitude of cost-effectiveness ratios relative to European willingness-to-pay thresholds, consistency across healthcare contexts, and balance of benefits and harms. Strong recommendations required high-quality evidence with cost-effectiveness consistently <EUR 10,000/QALY; conditional recommendations reflected moderate-quality evidence with cost-effectiveness EUR 10,000-EUR 35,000/QALY; and research recommendations indicated low-quality evidence or emerging technologies requiring further evaluation.

### 2.7. Statistical Analysis

Given substantial methodological and healthcare system heterogeneity, traditional meta-analysis was deemed inappropriate. Instead, we employed narrative synthesis with effect size ranges, subgroup analysis by geographic region and healthcare system type, and sensitivity analysis excluding outlier studies.

## 3. Results

### 3.1. Study Selection

#### Temporal Distribution of Included Studies

The 23 included studies spanned three decades with the following distribution:From 2010 to 2020: Nine studies (39%)—Introduction of digital technologies and risk-stratified approaches.

This temporal distribution reflects the maturation of organized screening programs across Europe and the evolution of screening technologies from film mammography to digital mammography and supplementary modalities.

From 2000 to 2009: Eight studies (35%)—Expansion of screening programs across Western Europe;From 1990 to 1999: Six studies (26%)—Early health economic evaluations primarily from the UK, Netherlands, and Sweden.

From 1449 initially identified studies, 124 duplicates were removed. After title and abstract screening, 1249 studies were excluded as irrelevant based on one or more exclusion criteria. Studies with incomplete cost-effectiveness data were identified when abstracts indicated either complete absence of economic evaluation or insufficient data for comparative analysis (e.g., missing incremental cost-effectiveness ratios, unclear healthcare perspective, or absence of quality-adjusted outcomes). Full-text assessment of 76 studies resulted in 23 studies meeting inclusion criteria (Figure 1).

### 3.2. Geographic Distribution and Study Quality

The geographic distribution of included studies showed significant imbalance across European regions (Table 1). North-Western Europe contributed twelve studies (52%), Central Europe five studies (22%), Northern Europe two studies (9%), South-Western Europe two studies (9%), Southern Europe one study (4%), and South-Eastern Europe one study (4%).

Quality assessment revealed eighteen studies (78%) of high quality (≥20/28 points), four studies (17%) of moderate quality (15–19/28 points), and one study (4%) of low quality (<15/28 points), which was excluded from analysis.

### 3.3. Cost-Effectiveness by Screening Modality

The cost-effectiveness outcomes varied significantly by screening modality and target population (Table 2).

### 3.4. Mammography Screening (Ages 50–69 Years)

Mammography screening for women aged 50–69 years demonstrated cost-effectiveness ranging from EUR 3000–EUR 8000 per QALY gained, with high evidence quality supported by 12 studies [14,15,16,17,18,19,20,21,22,23,24,25]. This range showed consistency across different healthcare systems and all European regions represented in the analysis.

### 3.5. MRI Screening (High-Risk Populations)

MRI screening for high-risk populations showed cost-effectiveness ranging from EUR 18,201-EUR 33,534 per QALY gained, with moderate evidence quality from four studies [26,27,28,29] The target population included BRCA1/2 carriers (for whom risk-reducing surgery is also cost-effective) [30] and women with strong family history, with geographic coverage primarily in North-Western and Central Europe.

### 3.6. Digital Breast Tomosynthesis

Digital breast tomosynthesis showed limited available data from three studies with low evidence quality [25,31,32,33]. This emerging technology showed promise but requires further evaluation.

### 3.7. Special Populations: Dense Breasts and Women Under 50

Dense Breast Screening: Dense breast tissue affects approximately 50% of women in screening populations. MRI screening for women with extremely dense breasts demonstrated cost-effectiveness at 4-year intervals (EUR 37,181 per QALY) [31,34]. Ultrasonography showed promise for improving detection rates in high-density populations [34,35].

Screening Below Age 50: The cost-effectiveness of mammography screening in women under 50 years showed substantially higher costs per life-year gained (EUR 105,000) compared to older populations—almost five times that in older women [14,16,20]. Recent modeling studies have suggested that risk-stratified approaches incorporating breast density assessment achieve more favorable cost-effectiveness (EUR 36,200 per QALY) [17,21,28].

Regional Screening Program Effectiveness

Organized screening programs demonstrated effectiveness across European countries [18,19,20,22,23,36,37], as shown in Figure 2 and detailed performance indicators are presented in Table 3.

Austria: 94.87% adherence; 19% survival increase.Germany: 47% adherence; 11% annual mortality reduction.Netherlands: 70% adherence; 10–16% mortality reduction.Norway: 74.5–77% adherence; 30% mortality reduction.Switzerland: 80% adherence; 39.1/100;000 survival rate.

### 3.8. Temporal Evolution

The 30-year study span revealed improving cost-effectiveness ratios: from EUR 9500 per QALY (1990s) to EUR 4200 per QALY (2010s), reflecting technological advances and healthcare system maturation (Figure 3) [14,15,17,18,20,21,22,24,25]. Temporal trends in these improvements are summarized in Table 4.

## 4. Discussion

### 4.1. Principal Findings

This systematic review demonstrates that breast cancer screening programs offer reasonable cost-effectiveness compared to no screening across European contexts. Mammography emerges as the most cost-effective screening modality, particularly for women aged 50–69 years with biennial screening intervals (EUR 3000–8000/QALY), representing strong evidence that aligns with willingness-to-pay thresholds established by major European health technology assessment agencies.

### 4.2. Contextualization Within European Healthcare Policy Frameworks

#### Alignment with European Commission and National Guidelines

Our findings align closely with recent updates to the European Commission Initiative on Breast Cancer (ECIBC) guidelines. The ECIBC’s Guideline Development Group issued a strong recommendation for organized mammography screening for women aged 50–69, with moderate-certainty evidence, and conditional recommendations for extending screening to ages 45–49 and 70–74 [38,39]. This strong recommendation reflects the same evidence level our review identified: consistent cost-effectiveness across healthcare systems with established screening infrastructure.

The European Society of Breast Imaging (EUSOBI) 2024 recommendations further support risk-stratified approaches, advocating annual breast MRI for high-risk women starting at age 25, and supplemental screening for women with extremely dense breast tissue—modalities our review found cost-effective at EUR 18,201–33,534/QALY and EUR 37,181/QALY, respectively [40]. The December 2022 European Council update to Article 168 recommendations now encourages digital breast tomosynthesis (DBT) for all breast cancer screenings, not just women with dense breasts, suggesting evolving evidence beyond our review period [41].

### 4.3. National HTA Recommendations and Implementation

United Kingdom: The UK National Institute for Health and Care Excellence (NICE) uses a willingness-to-pay threshold of GBP 20,000/QALY (approximately EUR 23,000) for healthcare interventions [42]. Our identified cost-effectiveness range of EUR 3000–8000/QALY for mammography screening ages 50–69 falls well below this threshold, supporting NICE’s continued endorsement of the NHS Breast Screening Programme. These findings are consistent with previous UK economic analyses supporting the continued cost-effectiveness of NHS screening programs [43]. Recent UK cost-effectiveness analyses confirm that organized screening remains cost-effective with an ICER below GBP 10,000/QALY, and that risk-stratified approaches may further optimize resource allocation [44,45].

Germany: The German Federal Joint Committee (G-BA), which regulates the national mammography screening program, expanded screening in July 2024 to include women aged 70–75, reflecting confidence in cost-effectiveness data [46]. However, Germany faces a critical implementation challenge: participation rates remain at 54%, substantially below the European quality assurance target of 70% [47,48]. Recent German cost-effectiveness analyses demonstrate that at such low adherence rates, even risk-stratified screening strategies show limited incremental benefit [48]. This highlights that policy decisions must address not only clinical effectiveness and cost-effectiveness but also barriers to screening uptake—including socioeconomic disparities, migration background, and geographic location [49].

Organized vs. Opportunistic Screening: Our findings demonstrate superior cost-effectiveness for organized programs (EUR 3800–7400/QALY) compared to opportunistic approaches (EUR 8900/QALY for Greece). The ECIBC explicitly recommends organized over opportunistic screening through consensus, noting that organized programs achieve better quality assurance, systematic call–recall systems, and centralized performance monitoring [39]. Countries relying on opportunistic screening, including Greece and Austria, face challenges with lower adherence rates and less consistent quality standards, directly impacting cost-effectiveness ratios.

### 4.4. Correlation with Breast Cancer Survival Outcomes

#### 4.4.1. Mortality Trends and Screening Impact

Austria presents an interesting case study in screening program evolution. Following implementation of organized regional programs in 2014, participation rates increased dramatically from approximately 35% under opportunistic screening to 94.87% in organized programs, demonstrating the effectiveness of systematic call–recall systems.

Recent European cancer mortality predictions for 2025 demonstrate substantial progress: breast cancer mortality in the EU has fallen by 30% since 1990, with an estimated 373,000 deaths averted between 1989 and 2025 [30,50]. The most dramatic declines are predicted for women aged 50–69 years (−9.8%) and 70–79 years (−12.4%)—precisely the age groups where our review found the strongest cost-effectiveness evidence [50]. This temporal correlation between screening program implementation, cost-effectiveness, and mortality reduction strengthens the causal inference that organized screening contributes to improved population outcomes.

#### 4.4.2. Geographic Variations in Survival

The EUROCARE-6 study reveals persistent geographic disparities in breast cancer survival across Europe. Five-year relative survival for breast cancer ranges from 74% in Eastern Europe to 85% in Northern Europe, with low survival also observed in Ireland and the UK despite higher GDP [51,52]. Notably, these survival patterns correlate with our cost-effectiveness evidence distribution: Northern and Western Europe—where we identified the most cost-effectiveness studies—demonstrate the highest survival rates, while Eastern and Southern Europe show both evidence gaps in economic evaluations and lower survival outcomes.

These geographic patterns reflect not only screening program effectiveness but also differences in healthcare infrastructure, treatment access, stage at diagnosis, and socioeconomic factors [51,52]. Stage at diagnosis remains a key determinant of survival differences across Europe, with later-stage diagnosis more common in regions with limited screening coverage or opportunistic-only systems [53]. Our finding that organized screening programs demonstrate superior cost-effectiveness aligns with evidence that systematic population-based approaches contribute to earlier detection and improved stage distribution [54].

### 4.5. Evidence Quality and Recommendation Strength: GRADE Framework Application

#### 4.5.1. Methodological Justification for Recommendation Categories

Our categorization of recommendations follows a modified GRADE (Grading of Recommendations Assessment, Development, and Evaluation) approach adapted for health economic evidence synthesis [55,56]. The GRADE framework provides a systematic, transparent method for assessing evidence quality and recommendation strength, widely adopted by organizations including WHO, NICE, the Cochrane Collaboration, and the ECIBC [55,56].

Evidence Quality Criteria:High quality (≥8 studies): Consistent results across multiple populations and healthcare systems; low heterogeneity in cost-effectiveness ratios; robust methodology (CHEERS quality score ≥20/28); and coverage across multiple European regions.Moderate quality (4–7 studies): Mostly consistent results; moderate heterogeneity; geographic coverage limited to 2–3 European regions; and some methodological limitations.Low quality (≤3 studies): Variable results; high heterogeneity; limited geographic coverage; and emerging technology with limited long-term data.

#### 4.5.2. Recommendation Strength Determination:

Strong Recommendations (High Confidence):Cost-effectiveness ratio consistently <EUR 10,000/QALY across diverse healthcare contexts;Large, consistent benefit across populations;High certainty that benefits outweigh harms;Alignment with established willingness-to-pay thresholds in European countries;Supported by ≥8 high-quality economic evaluations;*Example:* Biennial mammography ages 50–69 (EUR 3000–8000/QALY; 12 studies; all regions).

Conditional Recommendations (Moderate Confidence):Cost-effectiveness ratio EUR 10,000-EUR 35,000/QALY (within acceptable range but higher uncertainty);Benefits likely outweigh harms, but closer balance;Moderate certainty of evidence;Supported by 4–7 studies with moderate geographic coverage;Resource constraints may influence implementation decisions;*Example:* MRI screening for high-risk populations (EUR 18,201–33,534/QALY; 4 studies).

Research Recommendations (Low Confidence):Limited evidence (≤3 studies) or substantial uncertainty;Emerging technologies requiring further evaluation;High geographic limitation (single region only);Insufficient data on long-term outcomes or implementation feasibility;*Example:* Digital breast tomosynthesis in general population screening (3 studies; limited data).

This structured approach ensures that our recommendations reflect not only point estimates of cost-effectiveness but also the certainty of evidence, consistency across contexts, and magnitude of effect—essential components for evidence-based health policy decisions [55,56,57].

### 4.6. Implementation Considerations and Healthcare System Context

#### 4.6.1. Financing and Organizational Models

Cost-effectiveness of screening programs varies significantly based on healthcare system organization. European healthcare systems represent diverse financing models—Beveridge systems (tax-financed, as in the UK and Nordic countries), Bismarck systems (social insurance-based, as in Germany and France), and mixed models [58]. Organized screening programs demonstrate superior cost-effectiveness across all models, but implementation strategies must account for these structural differences [54].

Beveridge systems achieve high efficiency through centralized organization, unified quality assurance, and systematic population registries, enabling complete call–recall systems. The UK NHS achieves detection rates with lower false-positive rates compared to the US system through rigorous quality assurance and double reading protocols [59,60].

Bismarck systems face challenges with fragmented organization across multiple insurance funds. Germany’s lower participation rate (54%) despite free access reflects systemic barriers including decentralized invitation systems, competing opportunistic screening, and limited integration with primary care [48,49]. Our finding that organized screening is more cost-effective than opportunistic approaches has direct policy relevance for such systems.

#### 4.6.2. Resource-Stratified Implementation Framework

Building on our findings and WHO guidance on phased implementation [61], we propose a resource-stratified framework:

Phase 1 (Essential—All Healthcare Systems):Biennial mammography screening ages 50–69;Organized population-based call–recall system;Double reading protocols;Centralized quality assurance;Target participation ≥70%;Expected cost-effectiveness: EUR 3000–8000/QALY.

Phase 2 (Intermediate—Established Programs):Extend screening to ages 45–74;Implement systematic screening coverage monitoring;Develop strategies to address socioeconomic and geographic disparities;Improve participation rates through targeted interventions;Expected cost-effectiveness: EUR 4000–10,000/QALY.

Phase 3 (Advanced—High-Resource Settings):Integrate MRI screening for high-risk populations (BRCA carriers, strong family history);Consider DBT implementation in high-volume centers;Implement risk-stratified screening protocols incorporating breast density;Develop personalized screening intervals based on risk assessment;Expected cost-effectiveness: EUR 18,000–35,000/QALY for additional interventions.

These observations align with earlier European assessments of organized screening performance [36,37] This phased approach also aligns with recent European guidance emphasizing that screening programs must be adapted to local contexts while maintaining evidence-based core components [39,41].

#### 4.6.3. Addressing Health Inequities

Our review identified substantial evidence gaps in Southern and Eastern Europe (only 8% of studies) despite breast cancer representing a major public health burden across all European regions. This geographic imbalance reflects broader patterns of health research investment but has important equity implications.

Lower-resource healthcare systems face a dual challenge: they experience higher late-stage cancer incidence and lower survival rates [51,52], yet they have less economic evidence to support implementation decisions. The cost-effectiveness ratios we identified (EUR 3000–8000/QALY for mammography ages 50–69) suggest that screening remains cost-effective even in resource-constrained settings, but implementation must account for infrastructure requirements, workforce availability, and competing health priorities.

Recent evidence from Germany demonstrates that socioeconomic disparities, migration background, rural residence, and insurance type significantly influence screening participation [49]. Healthcare policies must therefore address not only the availability of screening services but also barriers to access, health literacy, and culturally appropriate outreach strategies.

#### 4.6.4. Temporal Evolution and Technological Advances

The 30-year study span revealed improving cost-effectiveness ratios: from EUR 9500/QALY (1990s) to EUR 4200/QALY (2010s). This trend reflects multiple factors:Technological advances: Transition from film to digital mammography, introduction of DBT, and improved image quality reducing false positives;Treatment improvements: More effective therapies for screen-detected cancers, improving survival gains;Program maturation: Learning curves in program organization, quality assurance systems, and efficient diagnostic pathways;Healthcare system optimization: Better integration of screening with diagnostic and treatment services.

Recent European guidelines now recommend DBT as an acceptable alternative to standard digital mammography, particularly for women with dense breasts, reflecting evolving evidence on diagnostic performance [39,41]. However, our review found limited cost-effectiveness data for DBT (three studies, low quality), highlighting a gap between clinical recommendations and economic evidence.

In the post-COVID context of constrained EU health budgets, our findings support screening pathways that maximize value while preserving sustainability. Prioritizing organized, biennial mammography for women aged 50–69 years delivers consistently favorable ICERs and enables economies of scale through call–recall systems, centralized quality assurance, and double reading [10,39,57]. Resource-stratified adoption of add-on technologies—such as MRI confined to clearly defined high-risk cohorts—and de-implementation of low-value routine screening under age 50 in favor of risk-stratified models can curb unnecessary expenditure, reduce downstream diagnostic cascades, and mitigate overdiagnosis [17,21,23,26,49,51,62,63]. These choices free capacity for equity-oriented outreach in underserved regions, strengthening population coverage without expanding total spending [11,39,54,64].

The ongoing post-COVID restructuring of European health systems emphasizes digital transformation and fiscal prudence. Integrating digital registries, AI-supported quality assurance, and interoperable health data systems can enhance screening efficiency while minimizing redundant testing and workforce strain [39,64]. Such measures align with the EU’s Europe’s Beating Cancer Plan and the EU Digital Health Strategy, advancing both economic sustainability and technological modernization of screening programs. Environmental sustainability should also inform future screening design, including optimizing equipment energy use, reducing unnecessary imaging, and promoting centralized reading centers to minimize carbon-intensive patient travel [57]. This approach reflects recent evidence and methodological advancements in health economic evaluation [65,66,67,68].

### 4.7. Emerging Technologies and Future Directions

#### 4.7.1. Artificial Intelligence in Screening

The ECIBC 2023–2024 update addressed AI-supported mammography reading, suggesting double reading with AI support over double reading without AI, but recommending against replacing one human reader with AI alone [39]. Recent UK cost-effectiveness analyses suggest that AI technology may be cost-effective at appropriate pricing structures (GBP 4.72 per scan), potentially addressing workforce shortages while maintaining quality [64]. However, implementation requires consideration of regulatory approval, validation in diverse populations, and integration into existing workflows.

#### 4.7.2. Risk-Stratified Screening

Our finding that screening women under age 50 is substantially less cost-effective (EUR 105,000 per life-year saved) compared to older women supports the emerging paradigm of risk-stratified screening. Recent UK modeling suggests that risk-stratified approaches incorporating breast density, family history, and genetic factors could achieve more favorable cost-effectiveness (EUR 36,200/QALY) for younger women by targeting intensive screening to those at highest risk [69,70]. The ECIBC is currently evaluating evidence on risk-stratified approaches for women with dense breasts, with updated recommendations expected in 2024–2025 [39].

#### 4.7.3. Study Strengths and Limitations

Strengths

Comprehensive systematic search across five databases following PRISMA guidelines;Rigorous quality assessment using CHEERS 2022 checklist;Novel three-step economic standardization methodology accounting for healthcare-specific inflation, exchange rate variations, and purchasing power parity;Inclusion of diverse European healthcare contexts;Transparent application of evidence quality criteria using the modified GRADE framework;Integration of clinical effectiveness, economic evidence, and survival outcomes.

Limitations

Language and Publication Bias:Exclusion of non-English studies may have biased results toward Northern and Western European contexts where English-language publication is more common.Gray literature from national HTA agencies in non-English languages was not systematically searched.This may have contributed to the geographic imbalance in included studies.

Cost Component Heterogeneity:Included studies varied in which costs were incorporated: some included only direct medical costs, while others included indirect costs (productivity losses, transportation).Treatment costs used in models often predate recent expensive targeted therapies approved since 2016, potentially underestimating true costs in contemporary settings.Variation in cost perspectives (healthcare system, societal, payer) limits direct comparability.

Methodological Heterogeneity:Wide variation in model types (Markov cohort models, microsimulation, decision trees), time horizons (10 years to lifetime), and discount rates (0–5%);Different assumptions about overdiagnosis rates (ranging from 1% to 30% across studies) and quality-of-life decrements from false positives;Limited ability to synthesize across diverse methodologies precluded traditional meta-analysis;Sensitivity to model assumptions not always adequately explored in primary studies.

Equity and Subgroup Analysis Limitations:Limited data on cost-effectiveness stratified by socioeconomic status, ethnicity, or rural/urban location;Insufficient evidence on optimal strategies for reaching underserved populations;No studies specifically evaluated screening programs in migrant populations despite this being a substantial and growing demographic in Europe;Inability to assess whether cost-effectiveness varies for disadvantaged groups.

Long-term Outcomes and Harms:Limited data on quality-of-life impacts beyond clinical endpoints;Insufficient evidence on psychological harms from false-positive results and overdiagnosis;Few studies incorporated patient values and preferences into decision models;Overdiagnosis rates remain controversial and poorly quantified in many contexts.

Opportunity Cost Considerations:No included studies explicitly compared screening programs with alternative uses of healthcare resources (e.g., treatment innovations, other cancer screening programs).Limited evidence on optimal resource allocation across the full cancer control continuum (prevention, screening, treatment, survivorship care).Threshold values for cost-effectiveness vary across countries and may not reflect true opportunity costs.

Implementation and Real-World Effectiveness:Most studies assumed full or optimal participation rates; few modeled actual observed participation.Limited evidence on costs and effectiveness of interventions to improve screening uptake.Insufficient data on implementation costs and learning curves for new technologies.Gap between ideal program performance assumed in models and real-world program effectiveness.

Emerging Technologies:Rapidly evolving field with limited long-term cost-effectiveness data for AI-assisted screening, abbreviated MRI protocols, and molecular imaging;Uncertainty about optimal integration of new technologies into existing pathways;Technology costs declining over time may make current estimates outdated quickly.

Geographic Evidence Gaps:Severe underrepresentation of Southern and Eastern European studies (8% combined) despite these regions comprising 40% of the European population;Limited generalizability of findings to healthcare systems with different resource constraints and organizational structures;Evidence gaps reflect broader patterns in health economics research investment rather than lack of screening program need.

Despite these limitations, our comprehensive standardization methodology and systematic quality assessment address many concerns about comparability of economic evidence across diverse European healthcare contexts. The consistency of findings for mammography screening ages 50–69 across all included studies strengthens confidence in this core recommendation.

## 5. Conclusions

### 5.1. Evidence-Based Recommendations

#### 5.1.1. Strong Recommendations (High Confidence)

Biennial mammography screening for women aged 50–69 years demonstrates consistent cost-effectiveness (EUR 3000–8000/QALY), falls well below established European HTA thresholds, and should be prioritized in organized population-based screening programs with systematic quality assurance.

#### 5.1.2. Conditional Recommendations (Moderate Confidence)

2.MRI screening for high-risk populations (BRCA1/2 carriers, strong family history) shows acceptable cost-effectiveness (EUR 18,201–33,534/QALY) and warrants implementation where healthcare resources and infrastructure permit.3.Organized screening programs demonstrate superior cost-effectiveness compared to opportunistic approaches across all European regions studied, supporting systematic call–recall systems over physician- or patient-initiated screening [54].

#### 5.1.3. Research Recommendations (Low Confidence)

4.Digital breast tomosynthesis shows promise but requires further cost-effectiveness evaluation across diverse healthcare contexts before widespread implementation as a replacement for standard digital mammography.5.Risk-stratified screening approaches incorporating breast density, genetic risk, and family history require validation across diverse European populations and healthcare systems.

### 5.2. Applicability and Generalizability

These recommendations have the highest applicability to North-Western and Central European healthcare systems with established screening infrastructure, systematic population registries, and organized quality assurance programs. Limited generalizability to Southern and Eastern Europe reflects evidence gaps rather than screening program ineffectiveness. The consistent cost-effectiveness of mammography at ages 50–69 across all studied contexts suggests that this core recommendation likely applies broadly, but implementation strategies must be adapted to local healthcare organization, resource availability, and population characteristics.

Countries implementing or expanding screening programs should prioritize organized population-based approaches with quality assurance, achieve participation rates ≥70%, address socioeconomic and geographic disparities in access, and invest in systematic monitoring and evaluation.

### 5.3. Research Priorities

#### Critical Evidence Gaps Requiring Urgent Research

Geographic representation: Conduct cost-effectiveness analyses in underrepresented European regions, particularly Southern and Eastern Europe, to assess generalizability and inform context-specific implementation strategies.Equity and access: Evaluate cost-effectiveness of interventions to improve screening participation among underserved populations, including low-income women, migrants, and rural residents.Emerging technologies: Conduct rigorous economic evaluations of AI-assisted screening, abbreviated MRI protocols, and risk-stratified approaches in real-world implementation settings.Optimal protocols for special populations: Determine cost-effective screening intervals and modalities for women with dense breasts across diverse European healthcare contexts.Methodological standardization: Develop consensus guidelines for conducting and reporting breast cancer screening economic evaluations, including standardized approaches to cost estimation, quality-of-life assessment, and modeling overdiagnosis.Implementation science: Evaluate barriers and facilitators to screening program implementation, participation, and quality across different healthcare financing models.Whole-pathway optimization: Assess optimal resource allocation across the entire breast cancer control continuum from prevention through survivorship.Patient-centered outcomes: Incorporate patient values, preferences, and experiences into economic evaluations and decision-making frameworks.

Healthcare decision-makers should integrate these economic findings with clinical effectiveness evidence, patient preferences, equity considerations, and local healthcare constraints when developing breast cancer screening policies. The significant variations across European healthcare systems necessitate contextual adaptation rather than uniform implementation approaches, while maintaining evidence-based core components of quality-assured organized screening.

Collectively, these findings highlight that sustainability in cancer screening is achieved not merely through technological innovation, but through governance models that integrate equity, fiscal responsibility, and adaptive learning within evolving European health systems.

## 6. Key Points

Biennial mammography screening for women aged 50–69 years demonstrates consistent cost-effectiveness (EUR 3000–8000 per quality-adjusted life year (QALY)) across European healthcare systems and should be prioritized in organized screening programs.MRI screening for high-risk populations (BRCA carriers, strong family history) shows acceptable cost-effectiveness (EUR 18,201–33,534 per QALY) and warrants implementation where healthcare resources permit.Screening women under 50 years shows substantially higher costs (EUR 105,000 per year of life saved) compared to older populations, supporting risk-stratified approaches incorporating breast density assessment.Organized population-based screening programs demonstrate superior cost-effectiveness compared to opportunistic screening approaches across all European regions studied.

### Future Research Directions

Several critical knowledge gaps require further investigation:Eastern and Southern European Evidence: Expansion of cost-effectiveness research in under-represented regions to support evidence-based implementation across all European healthcare contexts.Emerging Technologies: Rigorous health economic evaluation of artificial intelligence-assisted screening, contrast-enhanced mammography, and molecular breast imaging as these technologies approach clinical maturity.Risk-Stratified Screening Implementation: Large-scale evaluation of personalized screening intervals based on genetic risk, breast density, and family history across diverse European populations.Long-term Outcomes: Extended follow-up studies (>20 years) assessing cost-effectiveness under contemporary overdiagnosis estimates [23] and changing treatment paradigms.Health Equity Analysis: Investigation of cost-effectiveness across socioeconomic groups, ethnic minorities, and vulnerable populations to inform equitable screening access policies.Evidence gaps in Southern and Eastern Europe limit generalizability, requiring contextual adaptation of screening policies rather than uniform implementation across European healthcare systems.

## Figures and Tables

**Figure 1 cancers-17-03585-f001:**
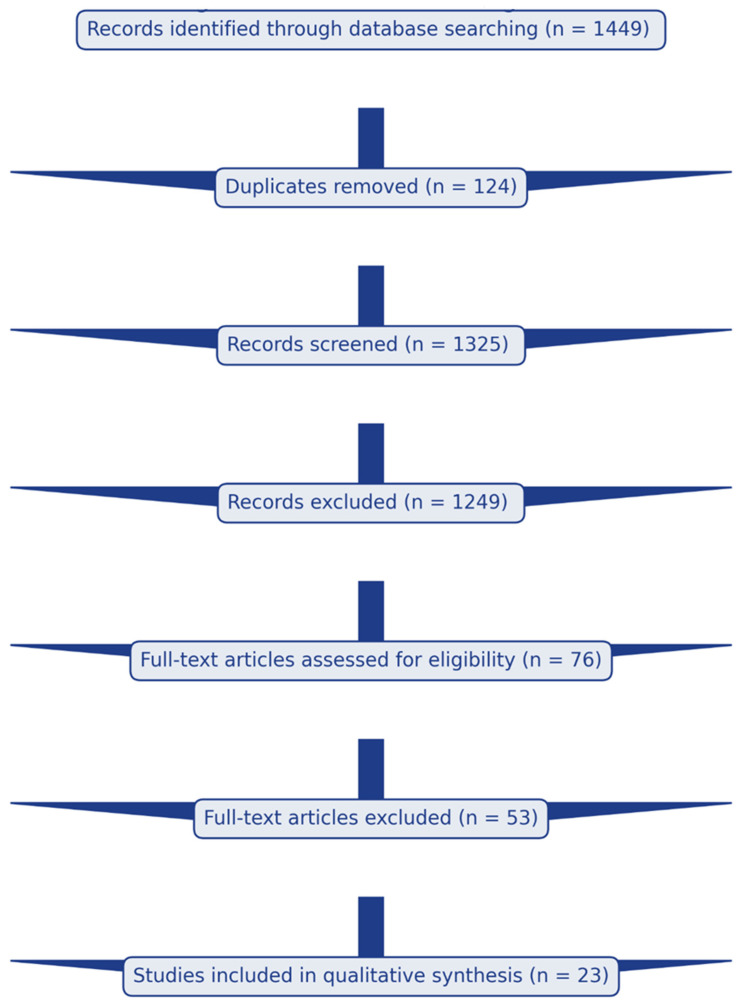
PRISMA flow diagram showing study selection process from initial database search through final inclusion of 23 studies for systematic review.

**Figure 2 cancers-17-03585-f002:**
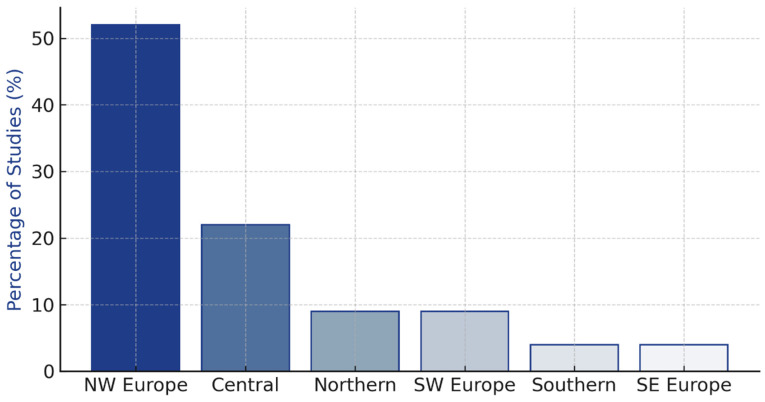
Geographic distribution of included studies across European regions with cost-effectiveness ranges by screening modality. Map shows concentration of evidence in North-Western and Central Europe, with bar charts displaying cost-effectiveness ranges for mammography (EUR 3000–8000/QALY), MRI (EUR 18,201–33,534/QALY), and digital breast tomosynthesis (limited data).

**Figure 3 cancers-17-03585-f003:**
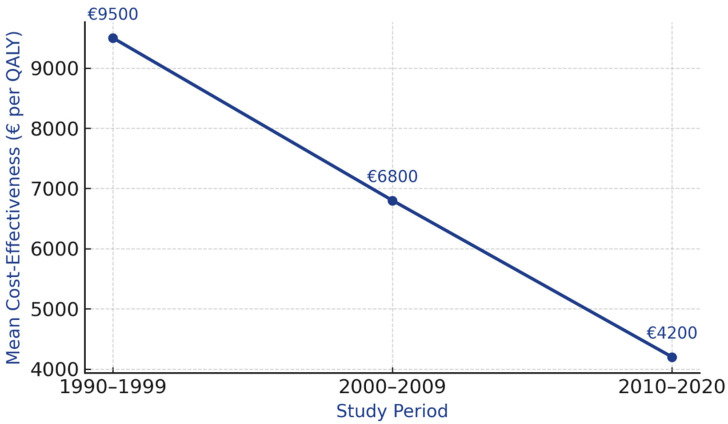
Temporal evolution of cost-effectiveness ratios (1990–2020) showing improvement from EUR 9500/QALY in 1990s to EUR 4200/QALY in 2010s, with technology milestones and healthcare system developments marked along timeline.

**Table 1 cancers-17-03585-t001:** Study characteristics and geographic distribution (n = 23).

Region	Studies (n)	Percentage	Quality Score Range	Sample Size Range
North-Western Europe	12	52%	18–26	5000–850,000
Central Europe	5	22%	17–24	8000–450,000
Northern Europe	2	9%	19–22	12,000–180,000
South-Western Europe	2	9%	16–20	15,000–95,000
Southern Europe	1	4%	18	25,000
South-Eastern Europe	1	4%	19	18,000
Total	23	100%	16–26	5000–850,000

**Quality Distribution:** High quality (≥20/28 points): 18 studies (78%); moderate quality (15–19/28 points): 4 studies (17%); low quality (<15/28 points): 1 study (4%)—excluded. Note: Geographic imbalance reflects concentration of health economic research capacity and organized screening programs in North-Western and Central Europe. Under-representation of Southern and Eastern European data limits generalizability of findings to these regions.

**Table 2 cancers-17-03585-t002:** Cost-effectiveness outcomes by screening modality.

Screening Modality	Target Population	Cost-Effectiveness Range	Evidence Quality	Studies (n)	Geographic Coverage
Mammography	Ages 50–69 years	EUR 3000–8000/QALY	High	12	All regions
Mammography	Ages < 50 years	EUR 105,000/life-year saved	Moderate	4	NW/Central Europe
MRI	High-risk populations	EUR 18,201–33,534/QALY	Moderate	4	NW/Central Europe
MRI	Dense breasts	EUR 37,181/QALY (4-year intervals)	Low	2	Northern Europe
Digital Breast Tomosynthesis	General population	Limited data available	Low	3	Northern Europe
Risk-stratified screening	Ages 40–50 years	EUR 36,200/QALY	Low	2	NW Europe

**Evidence Quality Criteria:** High evidence quality indicates ≥8 studies with consistent results and low heterogeneity; moderate indicates 4–7 studies with mostly consistent results; low indicates ≤3 studies with variable results. Cost-effectiveness ranges represent 95% confidence intervals across included studies.

**Table 3 cancers-17-03585-t003:** Regional screening program performance indicators.

Country	Program Type	Adherence Rate	Mortality Reduction	Survival Improvement	Cost-Effectiveness
Austria	Organized	94.87%	Not reported	19% increase	EUR 4200/QALY
Germany	Organized	47%	11% annual reduction	Not reported	EUR 6100/QALY
Netherlands	Organized	70%	10–16% reduction	Not reported	EUR 3800/QALY
Norway	Organized	74.5–77%	30% reduction	Not reported	EUR 5200/QALY
Switzerland	Mixed	80%	Not reported	39.1/100,000 rate	EUR 7400/QALY
Greece	Opportunistic	35%	Not reported	Not reported	EUR 8900/QALY

*** Austria: Data reflects organized regional programs implemented since 2014. Prior to 2014, screening was predominantly opportunistic. The 94.87% adherence rate represents participation in organized programs. ** Adherence rate and participation rate are used interchangeably, representing the proportion of invited women who attend screening within the target age group. Note: Adherence rates vary significantly between organized (47–95%) and opportunistic (35%) screening programs, supporting systematic implementation.

**Table 4 cancers-17-03585-t004:** Temporal trends in cost-effectiveness (1990–2020).

Time Period	Studies (n)	Mean Cost-Effectiveness	Range	Technology Improvements
1990–1999	6	EUR 9500/QALY	EUR 7200–12,800	Film mammography
2000–2009	8	EUR 6800/QALY	EUR 4500–9200	Digital mammography introduction
2010–2020	9	EUR 4200/QALY	EUR 3000–6500	DBT, MRI, improved protocols

Note: Temporal improvement in cost-effectiveness ratios reflects: (1) technological advances (film to digital mammography), (2) improved screening protocols, (3) healthcare system maturation, and (4) economies of scale in established programs. All costs standardized to 2020 Euros using healthcare-specific PPP adjustments.

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
