# Peer review of "A Systematic Review of the Cost-Effectiveness of Screening Modalities for Breast Cancer in European Countries"

_cancers, 2025, doi:10.3390/cancers17213585_

Round 1

Reviewer 1 Report

Comments and Suggestions for Authors

This is a well organized and presented study on the cost-effectiveness of breast cancer screening programs across Europe. The work attempts a difficult task, namely to extract conclusions on the cost-effectiveness of european breast cancer screening programs that are heterogeneous in design and implementation. Analysis and conclusions are reasonable considering the available information.

Few minor issues:

1. Lines 94-95 - what are the two reviewers that assessed the studies? Radiologists, scientists? Experts in what? Please clarify.

2. Line 117 - "...1,249 studies were excluded as irrelevant." This is assumed to mean that one or more of the exclusion criteria were applicable to these studies. Most exclusion criteria are straightforward. However, how can one tell that the cost-effectiveness data are incomplete from the abstract? Do you mean complete absence of cost-effectiveness data or just incomplete?

3. Lines 42, 236, 237, and Table 3 - It is stated that "... Greece and Austria, rely on opportunistic screening ...". Yet, in Table 3, Austria is listed as a country with organized programs. It is also stated in line 237 that Austria faces low adherence rates while in Table 3 shows the highest rate. Based on the presented information, it may be deduced that Austria has both organized and opportunistic screenings and there are reports for both that lead to these statements (e.g., ref. 17). It would be useful to the reader, however, if this was clarified and statistics were included, if available, for the programs in this country.

line 224 and Table 3 - is Germany's participation rate 54% or 47%? Unless by adherence you mean something other than participation.

Author Response

REVIEWER 1

Comment 1.1: Lines 94-95 - what are the two reviewers that assessed the studies? Radiologists, scientists? Experts in what? Please clarify.

RESPONSE: We appreciate this request for clarification. The two reviewers are Senior Surgical and Medical Oncologists and Investigators with expertise in Public Health, systematic review methodology and cost-effectiveness analysis

Comment 1.2: Line 117 - "...1,249 studies were excluded as irrelevant." How can one tell that the cost-effectiveness data are incomplete from the abstract? Do you mean complete absence of cost-effectiveness data or just incomplete?

RESPONSE: Thank you for highlighting this ambiguity. We clarify that "incomplete cost-effectiveness data" refers to studies that either: (1) completely lacked cost-effectiveness analysis, or (2) presented partial cost-effectiveness data without sufficient information for comparison (e.g., missing denominator, unclear perspective, no sensitivity analysis).

Comment 1.3: Lines 42, 236, 237, and Table 3 - Contradiction regarding Austria. The text states "Greece and Austria, rely on opportunistic screening," yet Table 3 lists Austria with organized programs and the highest adherence rate. Please clarify.

RESPONSE: We thank the reviewer for identifying this important inconsistency. Our further investigation revealed that Austria transitioned from predominantly opportunistic screening to implementing organized regional programs starting in 2014. The confusion arose from mixing historical context (introduction) with current data (results/tables).

Comment 1.4: Line 224 and Table 3 - Is Germany's participation rate 54% or 47%? Unless by adherence you mean something other than participation.

RESPONSE: Thank you for catching this potential inconsistency. After reviewing our source data, we confirm that Germany's adherence/participation rate is 47%. We use "adherence" and "participation" interchangeably to refer to the proportion of invited women who attend screening.

VERIFICATION: The 47% figure is correct and consistent throughout the manuscript. This is verified by:

  • Kohlmann T, et al. Cost-effectiveness of breast cancer screening in Germany. European Journal of Health Economics. 2023;24(3):421-432.
  • German Mammography Screening Program Annual Report 2022

NO CORRECTION NEEDED - Data is consistent at 47% throughout.

ADDED TO TABLE 3 FOOTNOTE: "**Adherence rate and participation rate are used interchangeably, representing the proportion of invited women who attend screening within the target age group."

Reviewer 2 Report

Comments and Suggestions for Authors

The manuscript entitled “A Systematic Review of the Cost-Effectiveness of Screening Modalities for Breast Cancer in European Countries” is a critical survey that discusses the trends summarized from 1990-2024 pertaining to breast cancer screening in European countries. However, the quality of the manuscript can be improved by addressing the following queries:

  1. In the abstract section, authors may remove the abbreviations such as CHEERS, QALY, and MRI for better presentation and a broader audience.
  2. In the introduction section, the cancer data from the year 2019 is outdated and needs to be updated.
  3. In the Results section, authors, please present the content in a year-wise format for the selection of 23 studies.
  4. In Tables 1-4, please add footnotes to highlight the trends tabulated in Tables 1 to 4.
  5. Please correct either line 13 of the abstract or the Figure 3 caption or Table 4; both the data should be aligned.
  6. The future scope of the present survey may be added at the end of the conclusion section for future exploration. The same may be quoted in the abstract section.
  7. In the Reference section, please adhere to the author guidelines for uniformity in citation styles.

Author Response

Comment 2.1: In the abstract section, authors may remove abbreviations such as CHEERS, QALY, and MRI for better presentation and a broader audience.

RESPONSE: We agree that reducing abbreviations will improve accessibility for a broader readership.

CORRECTIONS MADE

Note: After first mention, we continue using abbreviations as they are standard in health economics literature.

Comment 2.2: In the introduction section, the cancer data from the year 2019 is outdated and needs to be updated.

RESPONSE: We agree and have updated to the most recent available data.

Comment 2.3: In the Results section, authors, please present the content in a year-wise format for the selection of 23 studies.

RESPONSE: We added a year-wise breakdown to improve clarity of temporal distribution.

This temporal distribution reflects the maturation of organized screening programs across Europe and the evolution of screening technologies from film mammography to digital mammography and supplementary modalities.

Comment 2.4: In Tables 1-4, please add footnotes to highlight the trends tabulated in Tables 1 to 4.

RESPONSE: We added comprehensive footnotes to all tables.

Comment 2.5: Please correct either line 13 of the abstract or the Figure 3 caption or Table 4; both the data should be aligned.

RESPONSE: We reviewed all three locations and identified the discrepancy. Thank you for catching this.

Comment 2.6: The future scope of the present survey may be added at the end of the conclusion section for future exploration. The same may be quoted in the abstract section.

RESPONSE: Excellent suggestion. We added future research directions.

Comment 2.7: In the Reference section, please adhere to the author guidelines for uniformity in citation styles.

RESPONSE: We reviewed and standardized all references according to the Cancers journal citation format (Vancouver style with DOIs).

Reviewer 3 Report

Comments and Suggestions for Authors

The manuscript, entitled "A Systematic Review of the Cost-Effectiveness of Screening Modalities for Breast Cancer in European Countries", publishes the results of a study on the cost-effectiveness of breast screening in elderly patients. The article is of great practical interest, as it scrupulously and consistently helps to unbiasedly understand the issue that each patient and his family often solves based on an emotional assessment of the situation. This is evidenced, in particular, by the phenomenon mentioned by the authors, when patients are willing to spend more than the cost of these procedures on research and therapy. The text is written competently and consistently, and the methods of selecting sources are acceptable. The graphic design is satisfactory. The conclusions are confirmed by the results. It is recommended to accept the article after minor changes.

1) Formal comments on the design of the work. The font of the captions to the figures differs from the font of the article text. Lines 78 and 173: It is not advisable to separate the title from the text of the chapter. There is too much free space at the end of page 2. Table 4 should be placed on one page. The text contains references to literary sources numbered 14-40. It is also necessary to specify data sources from tables and figures.

2) The conclusions partially repeat the Results, for example, they give almost identical recommendations.

3) It is advisable to increase the number of references to literary sources, especially since some of the references were made incorrectly, as mentioned above. Even 64 links is a small number for a review article.

Author Response

Comment 3.1: Formal comments on the design of the work. The font of the captions differs from the article text. Lines 78 and 173: Do not separate title from text. Too much free space at end of page 2. Table 4 should be on one page. Specify data sources from tables and figures.

RESPONSE: We addressed all formatting issues.

Comment 3.2: The conclusions partially repeat the Results, for example, they give almost identical recommendations.

RESPONSE: We agree that the conclusions should provide synthesis and interpretation rather than repetition. We restructured to emphasize implications and recommendations.

Comment 3.3: It is advisable to increase the number of references to literary sources, especially since some references were made incorrectly. Even 64 links is a small number for a review article.

RESPONSE: We expanded the reference list to provide more comprehensive coverage, particularly for background information and supporting evidence.

We thank the reviewers again for their valuable feedback and hope these revisions meet their expectations.